# The Use of Neural Networks in Combination with Evolutionary Algorithms to Optimise the Copper Flotation Enrichment Process

**Dariusz Jamróz** [1] , **Tomasz Niedoba** [2,*] , **Paulina Pięta** [3] **and Agnieszka Surowiak** [2]

1   Faculty of Electrical Engineering, Automatics, Computer Science and Biomedical Engineering, Department of Applied Computer Science, AGH University of Science and Technology, al. Mickiewicza 30, 30-059 Kraków, Poland; jamroz@agh.edu.pl
2   Faculty of Mining and Geoengineering, Department of Environmental Engineering, AGH University of Science and Technology, al. Mickiewicza 30, 30-059 Kraków, Poland; asur@agh.edu.pl
3   JSW Innowacje S.A., ul. Paderewskiego 41, 40-282 Katowice, Poland; ppieta@jswinnowacje.pl
*   Correspondence: tniedoba@agh.edu.pl; Tel.: +48-12-617-20-56

**Abstract:** The paper presents a way of combining neural networks with evolutionary algorithms in order to find optimal parameters of the copper flotation enrichment process. The neural network was used in order to build a model describing the flotation process. The network learning was carried out with the use of samples from previous empirical measurements of the actual process. The model created in this way made it possible to find optimal parameters not only from among the measurement spaces, but also those that go beyond the measurements. Then, evolutionary algorithms were used in order to find optimal flotation parameters. The learned neural network previously described was used to calculate the criterion in the evolutionary algorithm.

**Keywords:** genetic algorithms; neural network; optimisation; evolutionary algorithms; flotation; copper ore

## 1. Introduction

Flotation is an enrichment method used to separate particles with different surface properties. During the process itself, particle-air bubble aggregates are formed and carried out into the flotation foam. However, copper-bearing minerals do not exhibit natural flotability, which results in the necessity to use certain substances for modifying their hydrophobicity. A thorough analysis of the entire copper ore enrichment process has led to the creation of a set of more than 100 parameters which have a significant effect on flotation results and have a significant effect on each other. The complex structure of the phenomenon under consideration leads to the selection of multidimensional methods of data analysis that are adjusted to the nature of the process [1–4]. Randomisation of flotation significantly hinders the creation of a model of the discussed phenomenon and its further optimisation with the use of classic tools, even using multidimensional methods of mathematical statistics [5].

Polish copper ore deposits are of the sedimentary type. They are characterised by varied mineralogical-petrographic and physico-chemical properties, complex chemical composition and close-grained mineralisation. An important property is the occurrence of three lithological types (carbonate, shale and sandstone), which determine high variability of susceptibility to enrichment processes [6–8]. A large variety of copper content in the deposit is described in Reference [9]. However, the correlation between the copper content in the deposit and the enrichment effect is not significant. In order to obtain high-quality copper concentrate, multi-stage flotation is used in Polish enrichment plants. The technological systems are adapted to enrich two types of the feed. Process lines are adapted

to enrich the carbonate and sandstone fractions separately. Therefore, the equipment installed in industrial conditions and separation parameters entered are different in these two cases [10]. An important factor in this process is the high level of release of copper-bearing minerals of different particle sizes depending on the lithological type [7,8]. Numerous studies indicate that the best flotation results are obtained for a feed with a particle size of 0.015–0.06 mm [11]. The analyses to date have shown that the distribution of sulphide minerals in copper ores is as follows [7,10]:

- Sandstone: 0.1–0.5 mm,
- Carbonate: 0.14–5 mm,
- Shale: 0.007–0.01 mm.

These values vary from mine to mine. Tables 1 and 2 show the characteristics of the feed entered into the technological system.

**Table 1.** Mineralogical composition of lithological types of Polish copper ores.

| Lithological Type of Copper Ore | Content of Selected Metals in Lithological Types | | Prevalent Copper-Bearing Minerals |
|---|---|---|---|
| carbonates | Cu (%) | 1.69 | chalcocite in combination with digenite, bornite, covellite and chalcopyrite |
| | Ag (g/t) | 54 | |
| shales | Cu (%) | 6.02 | chalcocite-bornite and bornite-chalcopyrite minerals |
| | Ag (g/t) | 188 | |
| sandstones | Cu (%) | 1.29 | bornite-chalcopyrite and chalcocite-bornite minerals |
| | Ag (g/t) | 30 | |

**Table 2.** Proportion of lithological types in the material mined in the KGHM Polska Miedź S.A. Mining Plants.

| Lithological Fraction | Branches of KGHM Polska Miedź S.A. | | | Total |
|---|---|---|---|---|
| | Lubin Branch | Polkowice-Sieroszowice Branch | Rudna Branch | |
| carbonates | 9.2 | 61.8 | 14.7 | 30.2 |
| shales | 11.8 | 9.7 | 5.9 | 8.7 |
| sandstones | 79.0 | 28.4 | 79.4 | 61.1 |

This study used the results of copper ore enrichment from the Polkowice-Sieroszowice mine. The feed has the highest proportion of the dolomite lithological type (about 62%) with mineral grain size below 100 μm. They are enriched with the use of the traditional system of flotation machines. These are pneumatic-mechanical machines. The performance of mechanical flotation machines for fine particles (<20 μm) is considered to be low due to the low probability of fine particle collisions and high turbulence levels. An increase in the likelihood of a particle-air bubble aggregate occurrence, where the size of the mineral particle is less than 0.1 mm, is possible by reducing the size of such bubbles in the flotation pulp. The Jameson cell (injector flotation machine) has a higher recovery potential in this area compared to the mechanical equipment. The reason for this is the difference in the distribution of bubble size in the flotation pulp [7,10,12–17].

## 2. The General Plan of the Experiment

In order to find the optimal flotation parameters, a number of empirical measurements were carried out. Each measurement consisted in establishing the conditions, i.e., the values constituting the process parameters: type of copper ore ($x_1$), particle size fraction ($x_2$), flotation time ($x_3$), collector type ($x_4$), collector dose ($x_5$), and in measuring the values resulting from the adopted parameters, i.e., the results of enrichment, i.e., copper content in the concentrate ($\beta_w$), copper content in tailings ($\vartheta_w$) and copper recovery in the concentrate ($\varepsilon_w$). However, it turned out that finding the optimal flotation parameters on the basis of the measurements carried out earlier is not a trivial problem.

Of course, the simplest solution would be to find one of the measurements carried out that would meet the assumed criterion to the greatest extent. To this end, it would be sufficient to check all samples in a loop and find the one for which the properly formulated criterion assumes the lowest value. However, in such a situation, the result would be limited to the measurements performed. In order to find optimal parameters not only from among the measurement spaces, a model describing the flotation process had to be constructed on the basis of the collected data. It was therefore necessary to create, on the basis of actual measurements, mapping which, for each sample, after providing flotation parameters at the mapping inputs, will result in the values at the mapping outputs that are closest to the enrichment effects present in the sample. In addition, such mapping should also work outside the samples, i.e., in the entire parameter space. There were 5 parameters and 3 enrichment values in each sample, so the mapping should transform the 5-dimensional space into the 3-dimensional space. Finding such mapping is not an easy issue; therefore, a decision to use the neural network to implement such a model was made.

A neural network learned with uncertainty back-propagation was used. The network had 5 inputs and 3 outputs. It consisted of 5 layers—4 initial layers consisted of 50 neurons, and the last one consisted of 3 neurons. A decision to use such a number of neurons due to the conducted experiments was made. With fewer neurons, the network could not learn all the samples correctly, and with too many neurons, the process of network learning took too long. The initial stage of the work consisted in the appropriate teaching of the neural network on the basis of samples from measurements, so that it would implement mapping describing the flotation process. Then, evolutionary algorithms were used in order to find the optimal flotation parameters.

One of the basic concepts in evolutionary algorithms is an individual. What an individual is depends on the specific problem to which the evolutionary algorithms are applied. In the problem analysed in this work, the individual is a sequence of 5 numbers. Each of these numbers corresponded to one of 5 flotation parameters adopted during the measurements. A population is a group of individuals. Operation of an evolutionary algorithm consists in changing the population of individuals in such a way that they change in the direction set by a particular criterion. At the end, the individual who best meets the applied criterion is selected. In the problem under consideration, the calculation of the criterion is possible due to the learned neural network. The evaluation of an individual consisted in providing 5 numbers which are the individual for 5 inputs of the learned neural network, and calculating criterion values on the basis of the values obtained for 3 network outputs.

## 3. Related Studies

The operation of evolutionary algorithms (GA) and artificial neural networks (NN) is based on selected mechanisms occurring in nature, respectively during the evolution of species and the operation of biological neural networks. It is undeniable that they are widely used in various fields of science. Both separately and in combination, they make it possible to obtain a high level of adjustment of the model to the empirical data, where the type of phenomenon under consideration does not really play a major role [18].

The possibility of combining neural networks with genetic algorithms has already been described [19–24]. In Reference [25], such a combination was used to optimise power systems through the introduction of a simple genetic algorithm as a teacher for the process of learning in advance (back-propagation of artificial neural networks). Other studies using the process/phenomenon optimisation with the use of a combination of a neural network and genetic algorithm in their methodologies are nearing different fields of science, although in fact, such an approach is not widely used due to the need to use advanced programming techniques [21–24].

Modelling and optimisation are an integral part of any technological system, process or phenomenon in the processing of mineral resources. It is caused by the need to increase production efficiency in the resource market. Many articles confirm the advantage of evolutionary algorithms [26–43] over the classical approach of mathematical statistics in this field of science.

Given the high flotation variability of non-ferrous metal ores, it can be concluded that both GA and NN are suitable tools to optimise this process. In this area, genetic algorithms have been used to design the feed flow in the technological system, maximise output and content of the useful ingredient in the foam product, determine the optimal number of flotation cycles and optimise the performance of the multi-stage flotation system [35,44–47]. The methodology used in the presented studies is limited to one of the methods discussed. This paper presents a method of optimisation of the flotation process with the use of a combination of these two techniques.

The data used in this work is multidimensional, so they can be treated as points in a multidimensional space. It suggests that all methods of qualitative analysis of this type of data and multidimensional spaces [48–60] can be used in their case, in particular to assess the correctness of these data and to observe some important features. In the case of multidimensional spaces, the possibility of analysing multidimensional continuous objects should also be mentioned [61].

## 4. Neural Networks

Neural networks learned with the use of the uncertainty back-propagation method were used in the work in order to implement the mapping, which is a flotation process model. In order to use such a network as a model, it should be taught in advance. At the beginning of this stage, the data should be scaled. The dataset was scaled to the range [−0.9, 0.9], that falls within the range of values obtained by the hyperbolic tangent function (−1, 1), which was used to calculate the values of neuron output. Since this function has been adopted, it is impossible to obtain values outside the range (−1, 1) at the network outputs. Initially, all the weight values for all the neurons should also be drawn.

The next points of the learning algorithm [48] should be implemented for each data sample obtained—this is a single learning iteration. Each sample consisted of the values $x_1$, $x_2$, $x_3$, $x_4$ and $x_5$, constituting parameters indicating: copper ore type, particle fraction, flotation time, collector type, collector dose and values $\beta_w$, $\vartheta_w$, $\varepsilon_w$, constituting enrichment results: copper content in concentrate, copper content in tailings and copper recovery in concentrate. The learning process should be repeated until the network is properly taught:

(1) For the next $w$-th data sample, the output values of all neurons in the first layer are calculated:

$$y_{1,j} = g\left(w_{1,j,0} + \sum_{k=1}^{n} w_{1,j,k} x_{k,w}\right), \tag{1}$$

where:

g—represents the assumed nonlinear function-hyperbolic tangent,

n—number of network inputs,

$y_{i,j}$—value of the neuron output placed in the $i$-th layer of the network at the $j$-th position ($i$ is 1 for neurons from the first layer),

$w_{i,j,k}$—represents the weight of the $k$-th neuron input placed in the $i$-th network layer at the $j$-th position,

$w_{i,j,0}$—denotes the constant component of a neuron placed in the $i$-th network layer at the $j$-th position,

$x_{k,w}$—represents the $k$-th parameter of the $w$-th data sample.

(2) Then, the values of neuron output from the other network layers are calculated, in the order from the second to the last layer:

$$y_{i,j} = g\left(w_{i,j,0} + \sum_{k=1}^{\text{size}(i-1)} w_{i,j,k} y_{i-1,k}\right), \tag{2}$$

where:

size $(i − 1)$ denotes the number of neurons in layer number $i − 1$, the other markings remain the same as in Equation (1).

Equations (1) and (2) are the standard equations being used by construction of one-directional neural networks. Thanks to such equations and assumed nonlinear function g, the neural network can be any projection of n-dimensional space into m-dimensional space. As function g, the hyperbolic tangent was accepted because it is a non-linear, differentiable and increasing function and its differential, necessary during the learning process, is easy to calculate. It is possible to achieve any adequacy by fitting such a network to create a model of projection of 5-dimensional input data space into the 3-dimensional space of results. This projection is a model describing the flotation process.

(3) The next step is the calculation of the network output errors, i.e., the differences between the values at the network outputs and the values that should be obtained. This difference is multiplied by the derivative of function *g* adopted in Equations (1) and (2), i.e., by the derivative of the hyperbolic tangent function, which results in obtaining:

$$\delta_{i,1} = \left(1 - y_{i,1}{}^2\right)\left(\beta_w - y_{i,1}\right), \tag{3}$$

$$\delta_{i,2} = \left(1 - y_{i,2}{}^2\right)\left(\beta_w - y_{i,2}\right), \tag{4}$$

$$\delta_{i,3} = \left(1 - y_{i,3}{}^2\right)\left(\beta_w - y_{i,3}\right), \tag{5}$$

where:

$\delta_{i,j}$—value of the calculated output error of the neuron placed in the *i*-th network layer at the *j*-th position (in this formula, *i* denotes the number of the last network layer),

$y_{i,j}$—output value of the *j*-th neuron from the *i*-th layer,

$\beta_w$, $\vartheta_w$, $\varepsilon_w$—enrichment results of the obtained *w*-th data sample (copper content in concentrate, copper content in tailings and copper recovery in concentrate).

(4) Then, the neuron output errors from other network layers are calculated, in the order from the penultimate layer to the first layer:

$$\delta_{i,j} = \left(1 - y_{i,j}{}^2\right) \sum_{k=1}^{size(j+1)} \left(\delta_{i+1,k} w_{i+1,k,j}\right), \tag{6}$$

where:

$\delta_{i,j}$—value of the calculated output error of the neuron placed in the *i*-th network layer at the *j*-th position,

$w_{i+1,k,j}$—represents the weight of the *j*-th input of the *k*-th neuron from the *i* + 1 layer,

size (*j* + 1)—denotes the number of neurons in layer *j* + 1,

$y_{i,j}$—output value of the *j*-th neuron from the *i*-th layer.

(5) Modification of the weights of all network neurons based on the previously calculated errors:

$$\widetilde{w}_{i,j,k} = w_{i,j,k} + \eta \delta_{i,j} y_{i-1,k}, \tag{7}$$

where:

$w_{i,j,k}$—represents the weight of the *k*-th input of the *j*-th neuron from the *i*-th layer,

$\widetilde{w}_{i,j,k}$—represents the weight $w_{i,j,k}$ after the change,

$\delta_{i,j}$—output error value of the *j*-th neuron from the *i*-th layer,

$y_{i-1,k}$—value of the *k*-th neuron output from the *i*-1-th layer,

$\eta$—learning speed (assumed value 0.001).

After the learning process is completed, the use of the network consists in calculating the values of the outputs based on the values of the inputs using Equations (1) and (2).

## 5. Evolutionary Algorithms

Evolutionary algorithms use mechanisms similar to those connected with the evolution of naturally occurring species. As already mentioned, one of the basic concepts in evolutionary algorithms is an individual. What an individual is depends on the specific problem to which the evolutionary algorithms are applied. In the problem analysed in this study, the individual is a sequence of 5 numbers. Each of these numbers corresponded to one of 5 flotation parameters obtained from the measurements. A population is a group of individuals. Operation of an evolutionary algorithm consists in changing the population of individuals in such a way that they change in the direction set by a particular criterion. The operation of the evolutionary algorithm created to solve the problem is as follows:

(1) The draw of the initial population consisting of $n = 5000$ individuals. Since each individual is 5 numbers, $5 \times 5000$ numbers should be drawn. In addition, the following limits were set on the range of volatility: $0 \le x_1 \le 2$, $10 \le x_2 \le 163$, $1 \le x_3 \le 30$, $0 \le x_4 \le 1$, $100 \le x_5 \le 200$. Furthermore, it is also assumed that $x_1$ and $x_4$ can only be integers. The allocated numbers $x_2$ indicate the middle points of the ranges. The adopted limits concerning the variability ranges also apply to all subsequent stages of the algorithm.

(2) Creating another population by merging:

- A copy of a previous population (consisting of $n$ individuals),
- $n$ individuals created as a result of a mutation of individuals from the previous population. The mutation consists in changing one of the numbers included in the individual to a random one or changing an arbitrary bit included in the individual to the opposite one.
- $n$ individuals created as a result of a crossover of individuals from the previous population. The crossover consists in creating a descendant (a new individual) in such a way that each number of which it consists is selected randomly from the first or second parent (parents are individuals from which the descendant was created). It is assumed that the crossover can also consist in creating a descendant in such a way that each bit of which the descendant consists is chosen randomly from the first or second parent.
- $n$ individuals created by searching a certain environment for the best individuals of the previous population. 10 modifications were created from a given individual. Each of the modifications consisted in changing one of the numbers included in this individual by some small value assumed. Since the adopted value can be added or subtracted and the individual consists of 5 numbers, 10 modifications were created from the given individual. Thanks to such a solution, the closest environment of the individual is examined in detail.

(3) (Selection—consists in population reduction. Only $n$ individuals are selected from among the $n = 4$ individuals created previously. It was assumed that the resulting population will consist of 40% of the best individuals (meeting the criterion to the highest degree), 20% of the worst individuals and the remainder of the randomly generated ones. Adding the worst individuals enables the whole population to emerge from local extrema. Adding randomly generated individuals results in the additional search in other space areas. In the problem under consideration, the calculation of the criterion is possible due to the learned neural network. The evaluation of an individual consisted in providing 5 numbers which are the individual ($x_1$, $x_2$, $x_3$, $x_4$, $x_5$) for 5 inputs of the learned neural network and calculating on the basis of the values obtained for 3 network outputs ($\beta$, $\vartheta$, $\varepsilon$) of the criterion values. It was assumed that we are looking for solutions resulting in obtaining maximum $\beta$, $\varepsilon$ and minimum $\vartheta$ values. Therefore, the minimisation of functions was adopted as the final criterion:

$$criterion = \vartheta - \beta - \varepsilon, \tag{8}$$

There is also a second version of the criterion, which adopts the above criterion with an additional restriction: $0.07 \le \beta \le 0.69$.

(4) If, in the course of evolution, the established number of generations has not yet been created and an interruption of evolution has not been requested, it is necessary to return to point 2, assuming the population established in point 3 as the current population.

After the completion of the algorithm presented above, an individual from the obtained population that meets the assumed criterion to the highest degree is a solution to the problem. This means that the numbers it consists of constitute the solution to the problem.

## 6. Experiment

The factorial experiment was carried out using a sample of copper ore with a nominal weight of 170 kg taken from the process line of the Polkowice Ore Enrichment Plant of KGHM Polska Miedź SA. Tests with the use of a Denver flotation machine were carried out at the AGH University of Technology (Faculty of Mining and Geoengineering, Department of Environmental Engineering and Mineral Processing) in Kraków, while studies concerning the copper ore enrichment in a Jameson cell were carried out in the laboratory of the Department of Mining Engineering, Faculty of Engineering in Dumlupınar Üniversitesi in Kütahya (Turkey). In order to implement the plan, a series of 72 tests in the Denver flotation machine and 12 tests in the Jameson cell was carried out. A full experimental plan was carried out for the first device; however, technological, transport and time constraints forced a reduction of the predicted scope for the second flotation machine. The flotation tests assumed the collective pre-flotation (30 min) and fractionated flotation cleaning the concentrate (30 min) for each sample. The concentrate was collected selectively after the following times: 1, 2, 4, 6, 9, 12, 17, 22 and 30 min. It allowed to analyse the kinetics of separation and investigate the effect of time on enrichment effects. The density of flotation pulp was maintained on a constant level of 300 g/dm$^3$ in the Denver machine, while the solid parts grade in the Jameson cell was equal to 2%. The agent Nasfroth was used in the amount of 50 g per 1 Mg of the feed as a frother during flotation tests. The final stage of the investigations was to determine the copper content in separation products by means of the XRF (X-ray fluorescence spectrometry) method. Before flotation started, the pulp was mixed for 3 min, then the collector was added, and the second stage of mixing occurred, which lasted 5 min. After this time, the frother was introduced in the dose of 50 g per 1 Mg of the feed and mixing continued for another 1 min.

Considering quality of the water, it can be stated that it differs during the investigations performed in laboratories in Poland and Turkey. However, the scale of devices, distance or the costs related to application of technological or distilled water made it impossible to unify this parameter. Distribution of air bubbles and pulp aeration are very important parameters but are hard to determine. In the Denver flotation machine, the impeller causes the turbulent motion conditions for appropriate mixing and aeration of the pulp. After the impeller, a stator is installed, which is a kind of calmer. Because of this, flowing conditions change and it is hard to determine these parameters for such multi-parameter flow. In the Jameson cell, another sort of pulp mixing occurs, which is basically without participation of gas phase. So, this is even harder to determine.

Therefore, a factorial experiment, in which 5 independent and 3 dependent variables (output) were determined, was conducted. The controllable parameters included: the lithological type of the copper ore, feed particle size, separation time, collector dose and type. The effects of the separation process, however, were evaluated by determining the following indicators: the copper content in concentrate, copper content in tailings and copper recovery in concentrate. It was necessary to repeat the process for each experimental level in order to verify the validity of the experiment course. The results of laboratory experiments were burdened with small and acceptable errors which were evaluated by means of a simple statistical measure of standard deviation.

Tables 3–6 present the exact experimental plan, which shows the variability range for each parameter. The time variable is not included in the table because each of the cleaning flotations was fractionated; hence the results for different enrichment times (1, 2, 4, 6, 9, 12, 17, 22 and 30 min) could be obtained during one experiment.

**Table 3.** The variability range of parameters of the flotation test planned and carried out according to the experimental plan (E—aqueous solution of ethyl sodium xanthate; Z—aqueous solution of isobutyl sodium xanthate)—Denver machine, sandstone copper ore.

| Test No. | Copper Ore Type | Particle Fraction (µm) | Collector Type | Collector Dose (g/Mg) |
|---|---|---|---|---|
| 1 | | 0–20 | E | 100 |
| 2 | | 0–20 | E | 150 |
| 3 | | 0–20 | Z | 100 |
| 4 | | 0–20 | Z | 150 |
| 5 | | 20–40 | E | 100 |
| 6 | | 20–40 | E | 150 |
| 7 | | 20–40 | Z | 100 |
| 8 | | 20–40 | Z | 150 |
| 9 | | 40–71 | E | 100 |
| 10 | | 40–71 | E | 150 |
| 11 | | 40–71 | Z | 100 |
| 12 | Sandstone | 40–71 | Z | 150 |
| 13 | copper ore | 71–100 | E | 100 |
| 14 | | 71–100 | E | 150 |
| 15 | | 71–100 | Z | 100 |
| 16 | | 71–100 | Z | 150 |
| 17 | | 100–125 | E | 100 |
| 18 | | 100–125 | E | 150 |
| 19 | | 100–125 | Z | 100 |
| 20 | | 100–125 | Z | 150 |
| 21 | | 125–200 | E | 100 |
| 22 | | 125–200 | E | 150 |
| 23 | | 125–200 | Z | 100 |
| 24 | | 125–200 | Z | 150 |

**Table 4.** The variability range of parameters of the flotation test planned and carried out according to the experimental plan (E—aqueous solution of ethyl sodium xanthate; Z—aqueous solution of isobutyl sodium xanthate)—Denver machine, dolomite copper ore.

| Test No. | Copper Ore Type | Particle Fraction (µm) | Collector Type | Collector Dose (g/Mg) |
|---|---|---|---|---|
| 25 | | 0–20 | E | 100 |
| 26 | | 0–20 | E | 150 |
| 27 | | 0–20 | Z | 100 |
| 28 | | 0–20 | Z | 150 |
| 29 | | 20–40 | E | 100 |
| 30 | | 20–40 | E | 150 |
| 31 | | 20–40 | Z | 100 |
| 32 | | 20–40 | Z | 150 |
| 33 | | 40–71 | E | 100 |
| 34 | | 40–71 | E | 150 |
| 35 | | 40–71 | Z | 100 |
| 36 | Dolomite | 40–71 | Z | 150 |
| 37 | copper ore | 71–100 | E | 100 |
| 38 | | 71–100 | E | 150 |
| 39 | | 71–100 | Z | 100 |
| 40 | | 71–100 | Z | 150 |
| 41 | | 100–125 | E | 100 |
| 42 | | 100–125 | E | 150 |
| 43 | | 100–125 | Z | 100 |
| 44 | | 100–125 | Z | 150 |
| 45 | | 125–200 | E | 100 |
| 46 | | 125–200 | E | 150 |
| 47 | | 125–200 | Z | 100 |
| 48 | | 125–200 | Z | 150 |

**Table 5.** The variability range of parameters of the flotation test planned and carried out according to the experimental plan (E—aqueous solution of ethyl sodium xanthate; Z—aqueous solution of isobutyl sodium xanthate)—Denver machine, shale copper ore.

| Test No. | Copper Ore Type | Particle Fraction (μm) | Collector Type | Collector Dose (g/Mg) |
|---|---|---|---|---|
| 49 | | 0–20 | E | 100 |
| 50 | | 0–20 | E | 150 |
| 51 | | 0–20 | Z | 100 |
| 52 | | 0–20 | Z | 150 |
| 53 | | 20–40 | E | 100 |
| 54 | | 20–40 | E | 150 |
| 55 | | 20–40 | Z | 100 |
| 56 | | 20–40 | Z | 150 |
| 57 | | 40–71 | E | 100 |
| 58 | | 40–71 | E | 150 |
| 59 | | 40–71 | Z | 100 |
| 60 | Shale | 40–71 | Z | 150 |
| 61 | copper ore | 71–100 | E | 100 |
| 62 | | 71–100 | E | 150 |
| 63 | | 71–100 | Z | 100 |
| 64 | | 71–100 | Z | 150 |
| 65 | | 100–125 | E | 100 |
| 66 | | 100–125 | E | 150 |
| 67 | | 100–125 | Z | 100 |
| 68 | | 100–125 | Z | 150 |
| 69 | | 125–200 | E | 100 |
| 70 | | 125–200 | E | 150 |
| 71 | | 125–200 | Z | 100 |
| 72 | | 125–200 | Z | 150 |

**Table 6.** The variability range of parameters of the flotation test planned and carried out according to the experimental plan (E—aqueous solution of ethyl sodium xanthate; Z—aqueous solution of isobutyl sodium xanthate)—Jameson cell, dolomite copper ore.

| Test No. | Copper Ore Type | Particle Fraction (μm) | Collector Type | Collector Dose (g/Mg) |
|---|---|---|---|---|
| 73 | | 0–25 | E | 100 |
| 74 | | 0–25 | E | 150 |
| 75 | | 0–25 | Z | 100 |
| 76 | | 0–25 | Z | 150 |
| 77 | | 25–45 | E | 100 |
| 78 | Dolomite | 25–45 | E | 150 |
| 79 | copper ore | 25–45 | Z | 100 |
| 80 | | 25–45 | Z | 150 |
| 81 | | 45–75 | E | 100 |
| 82 | | 45–75 | E | 150 |
| 83 | | 45–75 | Z | 100 |
| 84 | | 45–75 | Z | 150 |

A shale copper ore type is characterised with high content of sulfide copper minerals compared to other lithologic types. At the same time, it is a type of feed which is hard to beneficiate during flotation. In this work, the results were presented which were obtained only after one scavenger flotation stage, but the industrial practice is different. For such type of ores, 3–4 scavenger flotations are usually performed to obtain satisfying quality of concentrates. The assumed performance of multidimensional analyses was related to investigating various parameters. So, it was impossible to investigate 72 flotation tests with three repeats for each test because of time limitations.

On the basis of the evaluation of individual experiments, it was stated that the best results of enrichment were obtained during flotation of the sandstone copper ore type with particle size distribution of 40–71 μm, where an aqueous solution of ethyl sodium xanthate in the dose of 100 g/Mg was applied to the Denver flotation machine. The results are presented in Table 7.

**Table 7.** A balance of flotation parameters of copper ore with particle size distribution of 40–71 μm in the Denver flotation machine by means of an aqueous solution of ethyl sodium xanthate in the dose of 100 g/Mg.

| Flotation Products | $\gamma_i$ | $\Sigma\gamma_i$ | $\lambda_i$ | $\Sigma\beta_i$ | $\Sigma\vartheta_i$ | $\Sigma\varepsilon_i$ | $\Sigma\eta_i$ |
|---|---|---|---|---|---|---|---|
| Product no. 1 | 3.9% | 3.9% | 28.26% | 28.3% | 0.1% | 90.4% | 9.6% |
| Product no. 2 | 1.4% | 5.3% | 1.56% | 21.1% | 0.1% | 92.3% | 7.7% |
| Product no. 3 | 2.7% | 8.0% | 0.49% | 14.0% | 0.1% | 93.4% | 6.6% |
| Product no. 4 | 2.1% | 10.2% | 0.44% | 11.2% | 0.1% | 94.2% | 5.8% |
| Product no. 5 | 1.8% | 12.0% | 0.43% | 9.5% | 0.1% | 94.8% | 5.2% |
| Product no. 6 | 1.2% | 13.2% | 0.43% | 8.7% | 0.1% | 95.0% | 5.0% |
| Product no. 7 | 22% | 15.5% | 0.25% | 7.5% | 0.1% | 95.5% | 4.5% |
| Product no. 8 | 2.1% | 17.6% | 0.21% | 6.6% | 0.1% | 96.3% | 3.7% |
| Product no. 9 | 2.3% | 19.9% | 0.22% | 5.9% | 0.1% | 96.7% | 3.3% |
| Tailings | 80.1% | 100.0% | 0.05% | 1.2% | 0.00% | 100.0% | 0.0% |

In the case of enrichment of dolomite and shale copper ores in the Denver flotation machine, better results were obtained for the feed with particle size distribution of 20–40 μm by means of an aqueous solution of isobutyl sodium xanthate in doses of 150 g/Mg and 100 gMg, respectively. Considering the quality of flotation concentrates, these results were much worse than in the case of the sandstone copper ore type because of the material's worse floatability. Undeniably, sandstone copper ore enrichment was highly efficient, while the efficiency of shale copper ore flotation was much worse despite its high initial copper grade in the feed (Figure 1).

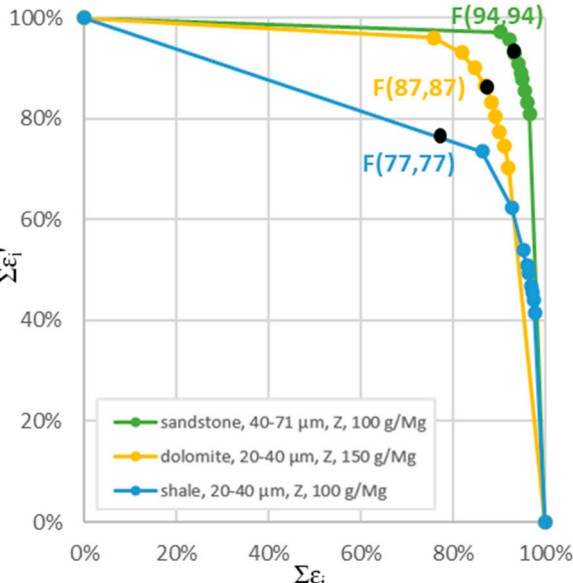

**Figure 1.** Fuerstenau's beneficiation curves for sandstone, dolomite and shale copper ore types, Denver flotation machine (E—aqueous solution of ethyl sodium xanthate; Z—aqueous solution of isobutyl sodium xanthate).

Having analysed the results from the Jameson cell, it was stated that the best results of enrichment were observed for the finest feed particle size distribution. That is why the information presented below was limited to the particle size fraction of 0–25 μm. The beneficiation curves indicate that

among the investigated cases, the best results of copper ore enrichment for particle size fraction of 0–25 μm were obtained for lower doses of collectors (100 g/Mg). An aqueous solution of isobutyl sodium xanthate ensured higher values of copper contents in concentrate, while ethyl sodium xanthate caused an increase in the copper recovery with a simultaneous decrease in the flotation concentrates' quality. However, it remained at the level comparable with the first reagent case.

The achievement of a high quality of flotation concentrates and maintenance of appropriate capacity of the process is still a challenge during enrichment of the feed with fine particle size distribution below 25 μm. The results of the tests performed in two flotation machines for the discussed feed particles' characteristics indicate that the separation process was different. The curves for the Jameson cell, presented on Figure 2, suggest that the recovery of gangue in tailings was very high, which clearly influences the high quality of froth concentrates. In the case of the Denver flotation machine, both copper recovery in concentrate ($\varepsilon$) and recovery of gangue in tailings ($\varepsilon'$) change relatively evenly. This means that the copper ore enrichment process conducted in the Denver machine is highly balanced because of the separation process evaluation factors mentioned above. Depending on the applied collectors, the optimal points, F, determined for the processes conducted in the Denver flotation machine were equal (79, 79) and (80, 80) respectively, for ethyl and isobutyl sodium xanthate. Because of the quality of flotation concentrates, the significant advantage in efficiency of fine particle fractions' enrichment can be observed in the Jameson cell.

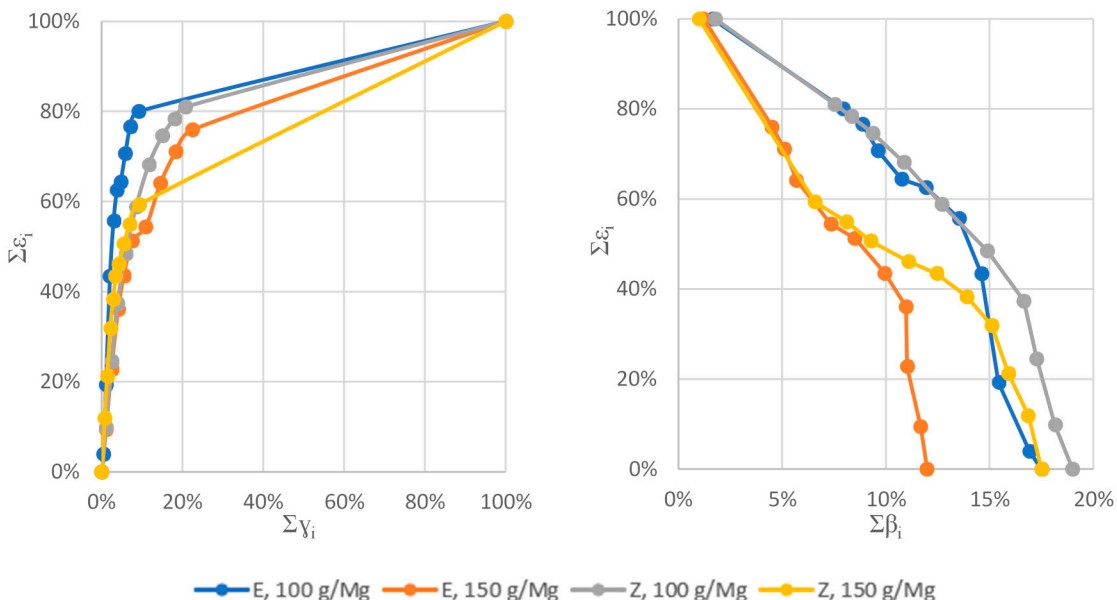

**Figure 2.** Mayer (left) and Halbich (right) curves for flotation of the dolomite copper ore type with particle size distribution of 0–25 μm by means of an aqueous solution of ethyl sodium xanthate (E) and isobutyl sodium xanthate (Z) in doses of 100 and 150 g/Mg in the Jameson cell.

The whole set of data are available in supplementary material in Tables A1–A4.

## 7. Results and Discussion

In order to carry out experiments to find the optimal flotation parameters, a computer program written in C++ language was created. The experimental data consisted of two parts: the first one, which concerned the results of enrichment in the Denver flotation machine and the second one, which concerned the results obtained in the Jameson cell.

Laboratory tests carried out with the use of the Denver flotation machine included 3240 cases. Tests concerning flotation enrichment of copper ore in the Jameson cell were limited to 540 cases. Above all, this was caused by technological differences between the flotation devices. Laboratory

tests were carried out for the feed with particle size of: 0–25, 25–45, 45–75 (µm), because the size distribution of air bubbles in the flotation chamber favours the separation of finer particles, which was confirmed by the previous experimental work. Therefore, it would be practically impossible to enrich copper ore in which particles above 71 µm would constitute the majority. Moreover, the transport restrictions resulted in the tests being carried out for only one type of lithological copper ore, namely dolomite ore. The choice was dictated by the fact that it is the predominant lithological type in the Polkowice-Sieroszowice mine from which the material for the research was obtained. The calculation process and data analysis were carried out in a similar way both for data obtained from the Denver flotation machine and for those obtained with the use of the Jameson cell. The limited number of experiments obtained using the Jameson cell resulted in a clear change in the accuracy of matching the model to empirical data. On their basis, the neural network determined a model for the flotation enrichment process of particular lithological types of Polish copper ore with selected process parameters. The model matching was assessed on the basis of the mean square error. In the process of network learning of the studied phenomenon, it was possible to compare the results of matching the model to empirical data. To illustrate the progress of change, the evaluation is summarised in Table 8.

**Table 8.** The evaluation of matching the model generated by the neural network depending on the number of repetitions used for two flotation machines.

| Number of Repetitions | Mean Square Error | |
|---|---|---|
| | Denver Flotation Machine | Jameson Cell |
| 1 | 0.032700 | 0.15642 |
| 50 | 0.013684 | 0.02103 |
| 100 | 0.011555 | 0.00753 |
| 200 | 0.010003 | 0.00419 |
| 500 | 0.008739 | 0.00287 |
| 1000 | 0.007937 | 0.00213 |
| 2000 | 0.007457 | 0.00129 |
| 3000 | 0.007222 | 0.00088 |

The evaluation of matching the model to the real conditions confirms the fact of increasing the accuracy with the number of repetitions in the learning process of the flotation phenomenon network, and at the same time, the time of obtaining the model is extended. Figure 3 shows two examples of 5 input parameters adopted for two tests conducted within 30 min. In order to illustrate the achieved accuracy of matching the model to the flotation parameters, the results obtained are presented in the diagrams in Table 9. It seems optimal to use 1000 repetitions, as the subsequent actions significantly increase the time of searching for a solution, while only slightly increasing the matching. In addition, with more repetitions, there is an increased risk of network overlearning which in these cases, needs to be further verified by dividing the data into a learning set and a verification set.

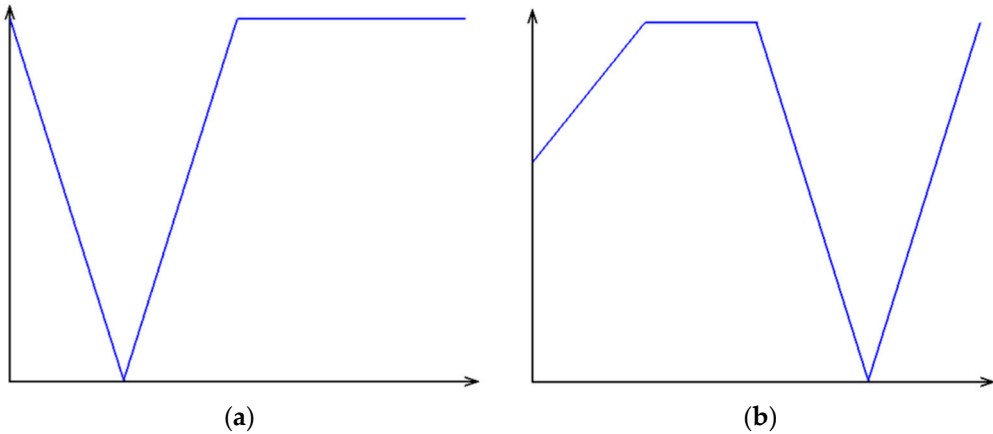

**Figure 3.** A comparison of 5 input parameters for test (**a**) 72 and (**b**) 84 carried out within 30 min.

**Table 9.** The graphical evaluation of matching the model of output parameters to empirical data according to the assumptions included in test 72 and 84 (red line—output values of the neural network obtained as a result of the learning process, blue line—empirical values).

| Number of Repetitions | Denver Flotation Machine (Test 72) | Jameson Cell (Test 84) |
| --- | --- | --- |
| 1 | | |
| 100 | | |
| 200 | | |
| 500 | | |
| 1000 | | |
| 2000 | | |

As a result of the neural network operation, a model of the flotation process of individual lithological types of Polish copper ore in the Denver flotation machine was created. Based on this model, the optimisation was done with the use of the evolutionary algorithm. The number of generations to be generated during the calculation process was verified. For example, it can be mentioned that when establishing one generation for the evolutionary algorithm, no satisfactory results were obtained. For the examined cases, a significant dispersion of the selected indicators' values could be observed. A gradual increase in the number of algorithm generations clarified the results. Tables 10 and 11 show the obtained optimal values of the copper ore enrichment process in the Denver flotation machine and in the Jameson cell, depending on the evolutionary algorithm advancement.

By analysing the obtained parameter values, it can be concluded that it seems sufficient to use 20 evolutionary algorithm generations for the flotation process optimisation since the results differ slightly above this value. The flotation process model created by NN extended the range of examined parameters beyond the measurement space and constituted the basis for the optimisation with the use of an evolutionary algorithm.

**Table 10.** A summary of the results of the copper ore flotation process optimisation in the Denver flotation machine taking the gradual development of the algorithm into account (E—aqueous solution of ethyl sodium xanthate).

| Number of Generations | Copper Ore Type | Particle Size (μm) | Cleaning Flotation Time (min) | Collector Type | Collector Dose (g/Mg) | $\beta$ (%) | $\vartheta$ (%) | $\varepsilon$ (%) |
|---|---|---|---|---|---|---|---|---|
| 5 | | 45–50 | 1.0–1.4 | E | 100–108 | 26–28 | 0.170–0.299 | 95.4–96.6 |
| 10 | | 46–48 | 1.15–1.42 | E | 104–106 | 27–28 | 0.208–0.285 | 97.0–98.5 |
| 15 | sandstone | 47 | 1.30–1.32 | E | 104–105 | 27.75 | 0.220–0.238 | 97.0–98.0 |
| 20 | | 47 | 1.30–1.31 | E | 104.98 | 27.72 | 0.225 | 97.60–97.61 |
| 25 | | 47 | 1.308–1.309 | E | 104.98 | 27.72 | 0.225 | 97.60–97.62 |
| 30 | | 47 | 1.308 | E | 104.98 | 27.72 | 0.225 | 97.60–97.63 |

**Table 11.** A summary of the results of the copper ore flotation process optimisation in the Jameson cell taking the gradual development of the algorithm into account (E—aqueous solution of ethyl sodium xanthate).

| Number of Generations | Copper Ore Type | Particle Size (μm) | Cleaning Flotation Time (min) | Collector Type | Collector Dose (g/Mg) | $\beta$ (%) | $\vartheta$ (%) | $\varepsilon$ (%) |
|---|---|---|---|---|---|---|---|---|
| 5 | | 0–25 | 30 | Z | 108.35 | 11.36 | 0.435 | 88.87 |
| 10 | | 0–25 | 30 | Z | 108.35 | 11.36 | 0.435 | 88.87 |
| 15 | dolomite | 0–25 | 30 | Z | 108.35 | 11.36 | 0.435 | 88.87 |
| 20 | | 0–25 | 30 | Z | 108.35 | 11.36 | 0.435 | 88.87 |
| 25 | | 0–25 | 30 | Z | 108.35 | 11.36 | 0.435 | 88.87 |
| 30 | | 0–25 | 30 | Z | 108.35 | 11.36 | 0.435 | 88.87 |

Comparing the results of the flotation process evaluation indicators ($\beta$, $\vartheta$, $\varepsilon$) for both devices, it can be concluded that the pneumatic-mechanical Denver flotation machine is characterised by higher efficiency in copper ore enrichment. However, it is misleading to some extent as the Jameson cell is a device dedicated to enrichment of the feed characterised by a finer particle size. At the same time, it should be noted that the Denver flotation machine is not able to ensure enrichment of copper ore with such particle characteristics. The clear difference in the flotability of the designated lithological types which favour the Denver flotation machine is also worth mentioning. The best results for the Denver flotation machine were obtained for easily enrichable sandstone copper ore.

Searching for optimal flotation parameters is a complex process and the results obtained from evolutionary algorithms and neural networks depend on the so-called "batch", i.e., experimental data on the basis of which the program learns the examined process. In order to check optimal conditions of the enrichment process for individual lithological types and particle fractions, the analysis was repeated while limiting the raw data to specific groups. In this way, information on input and output parameters of flotation enrichment of selected varieties of copper ore (Table 12) and particle size distribution of copper ore directed for separation in the Denver flotation machine (Table 13) was collected.

On the basis of the results presented in Table 12, it is possible to observe clear differences between the optimal separation process parameters of the existing copper ore types. On the basis of the results obtained from the genetic algorithm, it can be concluded that the highest quality of foam product is obtained during the flotation of sandstone ore (β = 24.85%) with copper loss (ηCu) in the tailings at the level of less than 5%. The quality of concentrates obtained as a result of the first cleaning flotation of dolomite ore is characterised by the copper content which is more than twice as low (10.91%). A similar trend is observed for the discussed indicator for shale ore (5.45%). With regard to the expected copper content in the tailings, the genetic algorithm estimated the range of 0.2–0.3%, which indicates the need to carry out further cleaning flotations in order to reduce copper losses. The discussed values of the cleaning flotation (I) performance evaluation indicators will be achievable with an appropriate level of experimental factors. GA has generated the following values: 37.23, 44.02 and 48.06 μm as the optimal size of the feed particles, for the dolomite, shale and sandstone types, respectively. The enrichment time needed to achieve the expected results for sandstone and dolomite copper ore is significantly shorter compared to the shale lithological type, which confirms the fact that the process of flotability is difficult for this lithological type.

**Table 12.** A summary of the optimisation results of the flotation process of particular lithological types from the Lubin-Głogów copper ore district in the Denver-type flotation machine (E—aqueous solution of ethyl sodium xanthate, Z—aqueous solution of isobutyl sodium xanthate, GA—genetic algorithms, NN—neural network).

| Flotation Process Parameters | Lithological Type of Copper Ore | | |
|:---:|:---:|:---:|:---:|
| | Dolomite | Sandstone | Shale |
| Number of data | 1080 | 1080 | 1080 |
| Number of GA generations | 30 | 30 | 30 |
| Number of NN repetitions | 5000 | 5000 | 1000 |
| Mean square error | 0.035986 | 0.005621 | 0.034646 |
| Optimal particle size (μm) | 37.23 | 48.06 | 44.02 |
| Flotation time (min) | 1.86 | 1.93 | 21.05 |
| Collector type | E | Z | E |
| Collector dose (g/Mg) | 100 | 100 | 131.52 |
| β (%) | 10.91 | 24.85 | 5.45 |
| ϑ (%) | 0.23 | 0.26 | 0.32 |
| ε (%) | 89.42 | 95.6 | 97.91 |

**Table 13.** A summary of the optimisation results of the flotation process of particular particle fractions from the Lubin-Głogów copper ore district in the Denver-type flotation machine (E—aqueous solution of ethyl sodium xanthate, Z—aqueous solution of isobutyl sodium xanthate).

| Flotation Process Parameters | Particle Fraction (μm) | | | | | |
|:---:|:---:|:---:|:---:|:---:|:---:|:---:|
| | 0–20 | 20–40 | 40–71 | 71–100 | 100–125 | 125–200 |
| Number of data | 540 | 540 | 540 | 540 | 540 | 540 |
| Number of GA generations | 30 | 30 | 30 | 30 | 30 | 30 |
| Number of NN repetitions | 1000 | 1000 | 1000 | 1000 | 1000 | 1000 |
| Mean square error | 0.014581 | 0.020541 | 0.021103 | 0.020013 | 0.024458 | 0.030046 |
| Lithological type of copper ore | | | sandstone | | | |
| Flotation time (min) | 22.81 | 30 | 1.316 | 30 | 23.24 | 30 |
| Collector type | Z | Z | Z | Z | E | Z |
| Collector dose (g/Mg) | 124.16 | 104.39 | 100 | 166.48 | 168.82 | 169.53 |
| β (%) | 10.93 | 14.27 | 22.33 | 5.87 | 7.66 | 7.98 |
| ϑ (%) | 0.058 | 0.07 | 0.1 | 0.05 | 0.11 | 0.05 |
| ε (%) | 98.57 | 98.22 | 92 | 96.18 | 95.83 | 90.9 |



When performing modelling and optimisation in the examined particle fractions, all lithological types were taken into account. The models obtained from NN were burdened with a mean square error of approximately 0.015–0.030. On their basis, optimisation, the results of which are summarised in Table 13, was carried out. Regardless of the particle size distribution of the flotation feed, the best results are obtained during the enrichment of sandstone copper ore. In the following particle fractions: 40–71, 20–40 and 0–20 (µm), the expected useful component content in the foam product is respectively: 22.33%, 14.27% and 10.93%, while the value of this indicator will reach the level of only 6–8% during enrichment of copper ore with particle size above 71 µm. Some discrepancies depending on the particle fraction being enriched can be observed based on the analysis of the optimisation results in terms of the expected time of separation of copper-bearing minerals from gangue rock in the first cleaning flotation. The data included in Table 13 indicate a high flotation speed of 40–71 µm particles. In the majority of the optimisation cases, an aqueous solution of isobutyl sodium xanthate in the following doses was indicated: 100–124 g of collector per 1 Mg of the feed with particle size distribution below 71 µm and 165–170 g/Mg for the other particle fractions.

The comparison of the results obtained in experimental tests conducted in the Denver flotation machine and optimal results generated by combined neural network and genetic algorithm method are presented in Tables 14–17.

**Table 14.** The results obtained during the analysis of all cases in the Denver flotation machine.

| Method | Copper ore Type | Particle Size (µm) | Cleaning Flotation Time [min] | Collector Type | Collector Dose (g/Mg) | $\beta$ (%) | $\vartheta$ (%) | $\varepsilon$ (%) |
|---|---|---|---|---|---|---|---|---|
| GA | Sandstone | 47 | 1.308 | E | 104.98 | 27.72 | 0.225 | 97.60–97.63 |
| LAB | | 40–71 | 1 | E | 100 | 28.26 | 0.12 | 90.55 |

where: GA—combined NN and GA method; LAB—laboratory test.

**Table 15.** The results obtained for the analysis of cases of all lithologic types separately in the Denver flotation machine.

| Flotation Process Parameters | Lithological Type of Copper Ore | | | | | |
|---|---|---|---|---|---|---|
| | **Dolomite** | | **Sandstone** | | **Shale** | |
| | LAB | GA | LAB | GA | LAB | GA |
| Method | LAB | GA | LAB | GA | LAB | GA |
| Optimal particle size (µm) | 20–40 | 37.23 | 40–71 | 48.06 | 40–71 | 44.02 |
| Flotation time (min) | 2.00 | 1.86 | 2.00 | 1.93 | 21.00 | 21.05 |
| Collector type | E | E | Z | Z | E | E |
| Collector dose (g/Mg) | 100.00 | 100.00 | 100.00 | 100.00 | 150.00 | 131.52 |
| $\beta$ (%) | 8.93 | 10.91 | 21.07 | 24.85 | 3.64 | 5.45 |
| $\vartheta$ (%) | 0.19 | 0.23 | 0.10 | 0.26 | 0.22 | 0.32 |
| $\varepsilon$ (%) | 88.03 | 89.42 | 92.39 | 95.6 | 98.45 | 97.91 |

**Table 16.** The results obtained for the analysis of cases of all particle size fractions separately in the Denver flotation machine.

| Flotation Process Parameters | Particle Fraction [µm] | | | | | | | | | | | |
|---|---|---|---|---|---|---|---|---|---|---|---|---|
| | **0–20** | | **20–40** | | **40–71** | | **71–100** | | **100–125** | | **125–200** | |
| | LAB | GA | LAB | GA | LAB | GA | LAB | GA | LAB | GA | LAB | GA |
| Method | LAB | GA | LAB | GA | LAB | GA | LAB | GA | LAB | GA | LAB | GA |
| Lithological type of copper ore | | | | | | sandstone | | | | | | |
| Flotation time (min) | 22.00 | 22.81 | 30.00 | 30.00 | 1.00 | 1.316 | 30.00 | 30.00 | 22.00 | 23.24 | 30.00 | 30.00 |
| Collector type | Z | Z | Z | Z | Z | Z | Z | Z | E | E | Z | Z |
| Collector dose (g/Mg) | 100.00 | 124.16 | 100.00 | 104.39 | 100.00 | 100.00 | 150.00 | 166.48 | 150.00 | 168.82 | 150.00 | 169.53 |
| $\beta$ (%) | 2.93 | 10.93 | 4.68 | 14.27 | 23.76 | 22.33 | 0.43 | 5.87 | 3.21 | 7.66 | 2.49 | 7.98 |
| $\vartheta$ (%) | 0.13 | 0.058 | 0.06 | 0.07 | 0.14 | 0.1 | 0.00 | 0.05 | 0.03 | 0.11 | 0.03 | 0.05 |
| $\varepsilon$ (%) | 95.61 | 98.57 | 97.08 | 98.22 | 87.92 | 92.00 | 100.0 | 96.18 | 94.73 | 95.83 | 89.33 | 90.9 |

**Table 17.** The results obtained for the analysis of all cases in the Jameson cell.

| Method | Copper Ore Type | Particle Size (μm) | Cleaning Flotation Time (min) | Collector Type | Collector Dose (g/Mg) | $\beta$ (%) | $\vartheta$ (%) | $\varepsilon$ (%) |
|--------|-----------------|---------------------|-------------------------------|----------------|------------------------|-------------|------------------|-------------------|
| AG | D | 0–25 | 30.00 | Z | 108.35 | 11.36 | 0.435 | 88.87 |
| LAB | D | 0.25 | 30.00 | Z | 100.00 | 7.55 | 0.42 | 81.00 |

The results presented in Tables 14–17 confirm that the optimisation results are converged, but they exceeded outside the range of investigations, establishing certain values of experimental parameters outside the range of data variation.

## 8. Conclusions

The paper showed a way of connecting neural networks with evolutionary algorithms. The neural network was used in order to build a model describing the flotation process. The network learning was carried out with the use of samples from previous empirical measurements of the actual process. The evolutionary algorithm, however, was used to find optimal flotation parameters based on the above model. This combination resulted in obtaining optimal results that go beyond the measurement space used to create the model.

The optimisation results of flotation enrichment of Polish copper ores in the Denver flotation machine and in the Jameson cell differ significantly. To some extent, this was influenced by the scope of experiments performed with the use of the second device. Performing a full series of flotation tests could change the final modelling and optimisation results. Different distribution of the size of air bubbles in the flotation chamber in both devices demonstrates the importance of this factor for the flotation results. Above all, the Jameson flotation machine enables the enrichment of fine particle fractions of copper ore, which is significantly hampered in the Denver flotation machine. Such a fine particle size of the feed is even a problem when enriching copper ores in pneumatic-mechanical machines.

By entering the full range of empirical data into the cleaning flotation optimisation software (I), the technological optimum was obtained for the following parameters:

- Flotation machine: Denver
- Feed: sandstone copper ore,
- Particle size: 40–71 μm ($d_{opt}$ = 47 μm),
- Cleaning flotation time: 1.308 min,
- Collector: aqueous solution of ethyl sodium xanthate,
- Collector dose: 105 g of collector per 1 Mg of the feed.

Maintaining the assumed level of the above-mentioned input parameters of the cleaning flotation process (I), it can be assumed that the content of copper-bearing minerals in the flotation concentrate will remain at the level of approximately 27.72%, the recovery will be approximately 97.61%, while the tailings will contain 0.225% of copper.

The cleaning flotation process (I) in the injector flotation machine in question will be most effective if the following parameters are used:

- Particle size: 0–25 μm,
- Cleaning flotation time: 30 min,
- Collector: aqueous solution of isobutyl sodium xanthate,
- Collector dose: 108.35 g of collector per 1 Mg of the feed.

The content of copper-bearing minerals in concentrate will remain at the level of approximately 11.36%, recovery will be approximately 88.87%, and tailings will contain 0.435% of copper.

Particle size and the characteristics of the feed being enriched are key factors that guarantee the correct and efficient flotation. The models obtained from NN were burdened with a mean square error of approximately 0.015–0.030. Regardless of the particle size of the feed, the sandstone lithological

type guarantees the best enrichment effects for the data range examined. The results of this study confirm the high efficiency of the adopted modelling methods in the description of the flotation process and in the search for the technological optimum for the separation of copper minerals from useless gangue rock. Of course, adding another parameter may improve the quality of the created flotation model. It can only adjust the obtained results because the projection of space of enlarged numbers of dimensions may have more optimal, or at least equally good extremes. The methods applied in this work can be transferred to investigations conducted at the industrial scale, but it requires a special way of conducting experimental research, ensuring that the conditions of measurements will be the same. The investigated factors can be related to characteristics of applied machines. Besides, the floated material can be also investigated because of its relation to the parameters in an analogical way to that considered in this paper.

**Supplementary Materials:** The following are available online at http://www.mdpi.com/2076-3417/10/9/3119/s1, Table A1. Averaged results of flotation tests of sandstone copper ore in Denver flotation machine (E—water solution of ethyl sodium xanthate; Z—water solution of isobutyl sodium xanthate). Table A2. Averaged results of flotation test of dolomite copper ore in Denver flotation machine (E—water solution of ethyl sodium xanthate; Z—water solution of isobutyl sodium xanthate). Table A3. Averaged results of flotation test of shale copper ore in Denver flotation machine (E—water solution of ethyl sodium xanthate; Z—water solution of isobutyl sodium xanthate). Table A4. Averaged results of flotation test of dolomite copper ore in Jameson cell (E—water solution of ethyl sodium xanthate; Z—water solution of isobutyl sodium xanthate).

**Author Contributions:** Conceptualization, D.J., T.N. and A.S.; methodology, D.J. and T.N.; software, D.J.; validation, D.J., T.N. and P.P.; formal analysis, T.N.; investigation, P.P.; resources, T.N. and A.S.; data curation, D.J. and T.N..; writing—original draft preparation, P.P.; writing—review and editing, D.J., T.N. and A.S.; visualization, D.J.; supervision, D.J., T.N. and A.S.; project administration, T.N.; funding acquisition, T.N. All authors have read and agreed to the published version of the manuscript.

**Funding:** This research received no external funding.

**Acknowledgments:** The paper is an effect of the realization of the statutory project no. 11.11.100.276.

**Conflicts of Interest:** The authors declare no conflict of interest.

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
