# Peer review of "The Use of Neural Networks in Combination with Evolutionary Algorithms to Optimise the Copper Flotation Enrichment Process"

_applsci, doi:10.3390/app10093119_

Round 1

Reviewer 1 Report

The authors endeavour to present techniques for optimising the processing of Polish copper ores using NN and evolutionary algorithms. This is a noble effort but I have the following objections to the publication of this manuscript:

In tables 3-6 there is a summary of lab tests completed but there is no information in the paper in regards to the ore preparation, handling and test conditions so. Furthermore, there is no discussion of the outcome of the test results prior to looking at advanced methods for predicting the outcomes.

I find it hard to believe that some of the conclusions presented could not have been reached with the data from the laboratory tests only. It is important that the raw test data be discussed and considered in conjunction with computer techniques. There is currently no evidence to suggest the computer techniques allowed for further optimisation beyond that which the raw test data presented.

Unfortunately, without proper control tests in place, I believe this manuscript will not do justice to the reputation of the journal. I, therefore, recommend it not be published but completely reviewed and resubmitted.

Reviewer 2 Report

Line 30. Missing residence time, air addition to flot cells, density of slurry, quality of water and etc. There is no study on particle size distribution (P80) as this can play immense role.

Line 525. 1 Mg? Is it 1 ton?

Overall, manuscript looks interesting, however, it considers only basic parameters of the flotation. There are some questions regarding implicating this in real plant. Because some factors were not considered in neural network as they can affect hugely in flotation like: cell designs, pump speed, opening and closing dart valves, inconsistency in the feed rate. I would suggest authors compare some results of this test in plant trial so can get more realistic data.

Reviewer 3 Report

The research presented in this paper is very interesting and is a hot topic.

  1. The authors may reorganize the structure of sections 1-3 to make them a whole part that aligns with a regular introduction section.
  2. In the article, authors mentioned some prior experiments work but without giving references, this may need to be clearer
  3. What are the reasons for using equations 1-2 in your modeling?
  4. Don’t you think the Cu grade is low for the shale type ore flotation is due to without properly controlling the chemisty? Like a depressant? Will you be able to improve the experimental and modeling results if you consider another reagent related parameter?
  5. I suggest you list all lab experimental results as appendix.
  6. In table 12, the results of flotation time for all different size fractions, do you think they are reasonable? Please refer some papers and explain it why you obtained these results.
